# Observation of the modification of quantum statistics of plasmonic systems

Chenglong You [1,6], Mingyuan Hong[1,6], Narayan Bhusal [1], Jinnan Chen[2], Mario A. Quiroz-Juárez[3], Joshua Fabre[1], Fatemeh Mostafavi[1], Junpeng Guo [2], Israel De Leon[4], Roberto de J. León-Montiel [5] & Omar S. Magaña-Loaiza [1✉]

For almost two decades, researchers have observed the preservation of the quantum statistical properties of bosons in a large variety of plasmonic systems. In addition, the possibility of preserving nonclassical correlations in light-matter interactions mediated by scattering among photons and plasmons stimulated the idea of the conservation of quantum statistics in plasmonic systems. It has also been assumed that similar dynamics underlie the conservation of the quantum fluctuations that define the nature of light sources. So far, plasmonic experiments have been performed in nanoscale systems in which complex multiparticle interactions are restrained. Here, we demonstrate that the quantum statistics of multiparticle systems are not always preserved in plasmonic platforms and report the observation of their modification. Moreover, we show that optical near fields provide additional scattering paths that can induce complex multiparticle interactions. Remarkably, the resulting multiparticle dynamics can, in turn, lead to the modification of the excitation mode of plasmonic systems. These observations are validated through the quantum theory of optical coherence for single- and multi-mode plasmonic systems. Our findings unveil the possibility of using multiparticle scattering to perform exquisite control of quantum plasmonic systems.

[1] Quantum Photonics Laboratory, Department of Physics and Astronomy, Louisiana State University, Baton Rouge, LA, USA. [2] Department of Electrical and Computer Engineering, University of Alabama in Huntsville, Huntsville, AL, USA. [3] Departamento de Física, Universidad Autónoma Metropolitana Unidad Iztapalapa, Ciudad México, Mexico. [4] School of Engineering and Sciences, Tecnologico de Monterrey, Monterrey, Nuevo Leon, Mexico. [5] Instituto de Ciencias Nucleares, Universidad Nacional Autónoma de México, Ciudad México, Mexico. [6] These authors contributed equally: Chenglong You, Mingyuan Hong. ✉email: maganaloaiza@lsu.edu

The observation of the plasmon-assisted transmission of entangled photons gave birth to the field of quantum plasmonics almost 20 years ago[1]. Years later, the coupling of single photons to collective charge oscillations at the interfaces between metals and dielectrics led to the generation of single surface plasmons[2]. These findings unveiled the possibility of exciting surface plasmons with quantum mechanical properties[3,4]. In addition, these experiments demonstrated the possibility of preserving the quantum properties of individual photons as they scatter into surface plasmons and vice versa[5–11]. This research stimulated the investigation of other exotic quantum plasmonic states[5,6,9,12,13]. Ever since, the preservation of the quantum statistical properties of plasmonic systems has constituted a well-accepted tenant of quantum plasmonics[3,4].

In the realm of quantum optics, the underlying statistical fluctuations of photons establish the nature of light sources[14,15]. These quantum fluctuations are associated to distinct excitation modes of the electromagnetic field that define quantum states of photons and plasmons[3,16,17]. In this regard, recent plasmonic experiments have demonstrated the preservation of quantum fluctuations while performing control of quantum interference and transduction of correlations in metallic nanostructures[6,8,10,11,18–24]. Indeed, the idea behind the conservation of quantum statistics of plasmonic systems results from the simple single-particle dynamics supported by the plasmonic nanostructures and waveguides used in previous experiments[6,8,10,11,18,20–24]. Despite the dissipative nature of plasmonic fields, the additional interference paths provided by optical near fields have enabled the harnessing of quantum correlations and the manipulation of spatial coherence[18,21,25–27]. So far, this exquisite degree of control has been assumed independent of the excitation mode of the interacting particles in a plasmonic system[3]. Moreover, the quantum fluctuations of plasmonic systems have been considered independent of other properties such as polarization, temporal, and spatial coherence[5,21,25–27]. Hitherto, physicists and engineers have been relying on these assumptions to develop plasmonic systems for quantum control, sensing, and information processing[3,4,16,18,23,24,28].

Previous research has stimulated the idea that the quantum statistical properties of bosons are always preserved in plasmonic systems[3–7,9]. Here, we demonstrate that this is not a universal behavior and consequently the quantum statistical fluctuations of a physical system can be modified in plasmonic structures. We reveal that scattering among photons and plasmons induces multiparticle interference effects that can lead to the modification of the excitation mode of plasmonic systems. Remarkably, the multiparticle dynamics that take place in plasmonic structures can be controlled through the strength of the optical near fields in their vicinity. We also show that plasmonic platforms enable the coupling of additional properties of photons to the excitation mode of multiparticle systems. More specifically, we demonstrate that changes in the spatial coherence of a plasmonic system can induce modifications in the quantum statistics of a bosonic field. Given the enormous interest in multimode plasmonic platforms for information processing[8,9,11,18], we generalize our single-mode observations to a more complex system comprising two independent multiphoton systems. We validate our experimental results through the quantum theory of optical coherence[14]. The possibility of controlling the underlying quantum fluctuations of multiparticle systems has important implications for practical quantum plasmonic devices[3,4].

## Results and discussion

**Theory**. We now introduce a theoretical model to describe the global dynamics experienced by a multiphoton system as it scatters into surface plasmons and vice versa. Interestingly, these photon–plasmon interactions can modify the quantum fluctuations that define the nature of a physical system[14,16,17]. As illustrated in Fig. 1a, the multiparticle scattering processes that take place in plasmonic structures can be controlled through the strength of the confined near fields in their vicinity. The near-field strength defines the probability of inducing individual phase jumps through scattering[11]. These individual phases establish different conditions for the resulting multiparticle dynamics of the photonic–plasmonic system.

We investigate the possibility of modifying quantum statistics in the plasmonic structure shown in Fig. 1b. The scattering processes in the vicinity of the multi-slit structure lead to additional interference paths that affect the quantum statistics of multiparticle systems[18,27]. The dynamics can be induced through either propagating or non-propagating near fields. Consequently, one could study this phenomenon using localized or propagating surface plasmons[29]. In particular, for this study, we focus on the propagating plasmonic fields supported by the structure. The gold structure in Fig. 1b consists of two slits aligned along the $y$-direction (see the "Methods" section). The structure is designed to excite plasmons when it is illuminated with thermal photons polarized along the $x$-direction[26,27]. For simplicity, we will refer to light polarized in the $x$- and $y$-direction as horizontally $|H\rangle$ and vertically $|V\rangle$ polarized light, respectively. Our polarization-sensitive plasmonic structure directs a fraction of the horizontally polarized photons to the second slit when the first slit is illuminated with diagonally $|D\rangle$ polarized photons. As depicted in Fig. 1b, this effect is used to manipulate the quantum statistics of a mixture of photons from independent multiphoton systems[30,31].

The modification of quantum statistics induced by the scattering paths in Fig. 1b can be understood through the Glauber-Sudarshan theory of coherence[14,32]. For this purpose, we first define the $P$ function associated to the field produced by the indistinguishable scattering between the two independently-generated, horizontally polarized fields. These represent either photons or plasmons emerging through each of the slits.

$$P_{Pl}(\alpha) = \int P_1(\alpha - \alpha')P_2(\alpha')d^2\alpha'. \qquad (1)$$

The $P$ function for a thermal light field is given by $P_i(\alpha) = (\pi\bar{n}_i)^{-1}\exp(-|\alpha|^2/\bar{n}_i)$. Here, $\alpha$ describes the complex amplitude as defined for coherent states $|\alpha\rangle$. The mean particle number of the two modes is represented by $\bar{n}_1 = \eta\bar{n}_S$ and $\bar{n}_2 = \bar{n}_{Pl}$. Moreover, the mean photon number of the initial illuminating photons is represented by $\bar{n}_S$, whereas $\bar{n}_{Pl}$ describes the mean photon number of scattered plasmonic fields. The parameter $\eta$ is defined as $\cos^2\theta$. The polarization angle, $\theta$, of the illuminating photons is defined with respect to the vertical axis. Note that the photonic modes in Eq. (1) can be produced by independent multiphoton systems. Furthermore, we make use of the coherent state basis to represent the state of the combined field as $\rho_{Pl} = \int P_{Pl}(\alpha)|\alpha\rangle\langle\alpha|d^2\alpha$[32]. This expression enables us to obtain the probability distribution $p_{Pl}(n) = \langle n|\rho_{Pl}|n\rangle$ for the scattered photons and plasmons with horizontal polarization. We can then write the combined number distribution for the multiparticle system at the detector as $p_{det}(n) = \sum_{m=0}^{n} p_{Pl}(n-m)p_{Ph}(m)$. The distribution $p_{Ph}(m)$ accounts for the vertical polarization component of the illuminating multiphoton systems. Thus, we can describe the final photon-number distribution after the plasmonic structure as (see Supplementary Note 1)

$$p_{det}(n) = \sum_{m=0}^{n} \frac{(\bar{n}_{Pl} + \eta\bar{n}_S)^{n-m}\left[(1-\eta)\bar{n}_S\right]^m}{(\bar{n}_{Pl} + \eta\bar{n}_S + 1)^{n-m+1}\left[(1-\eta)\bar{n}_S + 1\right]^{m+1}}. \qquad (2)$$

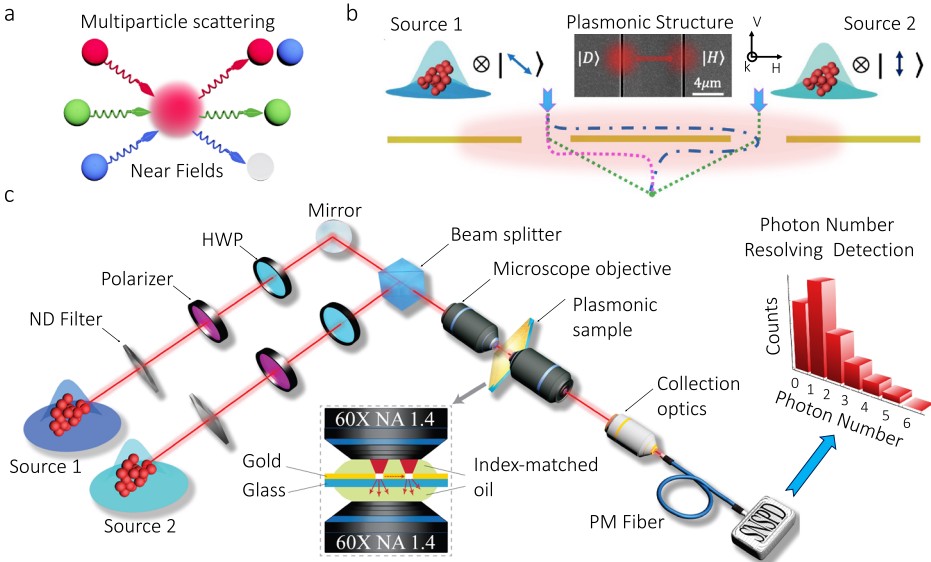

**Fig. 1 Multiparticle scattering in plasmonic systems.** The diagram in (**a**) illustrates the concept of multiparticle scattering mediated by optical near fields. The additional interference paths induced by confined near fields lead to the modification of the quantum statistics of plasmonic systems. This idea is implemented through the plasmonic structure shown in (**b**). The dotted lines represent additional scattering paths induced by confined optical fields in the plasmonic structure[27]. Our metallic structure consists of a 110-nm-thick gold film with slit patterns. The width and length of each slit are 200 nm and 40 μm, respectively. The slits are separated by 9.05 μm. The fabricated sample is illuminated by either one or two thermal sources of light with specific polarizations. The strength of the plasmonic near fields is controlled through the polarization of the illuminating photons. The plasmonic near fields are only excited with photons polarized along the horizontal direction. The experimental setup for the observation of the modification of quantum statistics in plasmonic systems is shown in (**c**). We prepare either one or two independent thermal multiphoton states with specific polarizations. The polarization state of each of the multiphoton systems is individually controlled by a polarizer (Pol) and half-wave plate (HWP). The two multiphoton states are injected into a beam splitter (BS) and then focused onto the gold sample through an oil-immersion objective. The refractive index of the immersion oil matches that of the glass substrate creating a symmetric index environment around the gold film. The transmitted photons are collected with another oil-immersion objective. We measure the photon statistics in the far field using a superconducting nanowire single-photon detector (SNSPD) that is used to perform photon-number-resolving detection[33,34].

Note that the quantum statistical properties of the photons scattered from the sample are defined by the strength of the plasmonic near fields $\bar{n}_{Pl}$. As illustrated in Fig. 1a, the probability function in Eq. (2) demonstrates the possibility of modifying the quantum statistics of photonic–plasmonic systems. It is worth remarking that Eq. (2) is valid only when the two sources, i.e., the two slits, are active and contribute to the combined field measured by the detector.

**Experimental results and discussions.** We first explore the modification of quantum statistics in a plasmonic double-slit structure illuminated by a thermal multiphoton system[30,31]. The experimental setup is depicted in Fig. 1c. This allows us to focus thermal photons onto a single slit and measure the far-field spatial profile and the quantum statistics of the transmitted photons. As shown in Fig. 2, we perform multiple measurements corresponding to different polarization angles of the illuminating photons. As expected, the spatial profile of the transmitted photons does not show interference fringes when the photons are transmitted by the single slit (see Fig. 2a). However, the excitation of plasmonic fields increases the spatial coherence of the scattered photons. As indicated by Fig. 2b–d, the increased spatial coherence leads to the formation of interference structures. This effect has been observed multiple times[8,21,26,27]. Nonetheless, it had been assumed independent of the quantum statistics of the hybrid photonic–plasmonic system[3,4]. However, as demonstrated by the probability distributions from Fig. 2e–h, the modification of spatial coherence is indeed accompanied by a modification of the quantum fluctuations of the plasmonic system. This effect had not been observed before as measurement devices used in

previous experiments were insensitive to the multiparticle dynamics supported by this kind of plasmonic structures[6,9,11,18,21,24,26,27]. In our case, the modification of the quantum statistics, mediated by multiparticle scattering, is captured through a series of photon-number-resolving measurements[33,34]. In our experiment, the mean photon number of the photonic source $\bar{n}_s$ is three times the mean particle number of the plasmonic field $\bar{n}_{Pl}$. The theoretical predictions for the photon-number distributions in Fig. 2 were obtained using Eq. (2) for a situation in which $\bar{n}_s = 3\bar{n}_{Pl}$.

The sub-thermal photon-number distribution shown in Fig. 2f demonstrates that the strong confinement of plasmonic fields can induce anti-thermalization effects[35]. Here, the scattering among photons and plasmons attenuates the chaotic fluctuations of the injected multiphoton system, characterized by a thermal photon-number distribution, as indicated by Fig. 2e. Conversely, the transition in the photon-number distribution shown from Fig. 2g–h is mediated by a thermalization effect. In this case, the individual phase jumps induced by photon-plasmon scattering increases the chaotic fluctuations of the multiparticle system[11], leading to the thermal state in Fig. 2h. As shown in Fig. 3, the quantum statistics of the photonic–plasmonic system show an important dependence on the strength of the optical near fields surrounding the plasmonic structure. The photon-number-distribution dependence on the polarization angle of the illuminating photons is quantified through the degree of second-order coherence $g^{(2)}$ in Fig. 3[33]. We use the measured photon statistics to evaluate $g^{(2)}$ defined as $g^{(2)}(\tau) = 1 + \left(\langle (\Delta \hat{n})^2 \rangle - \langle \hat{n} \rangle\right)/\langle \hat{n} \rangle^2$[36]. This coherence function is independent of time for single-mode fields[15]. Furthermore, we point out that while Eq. (2) depends on the brightness of the

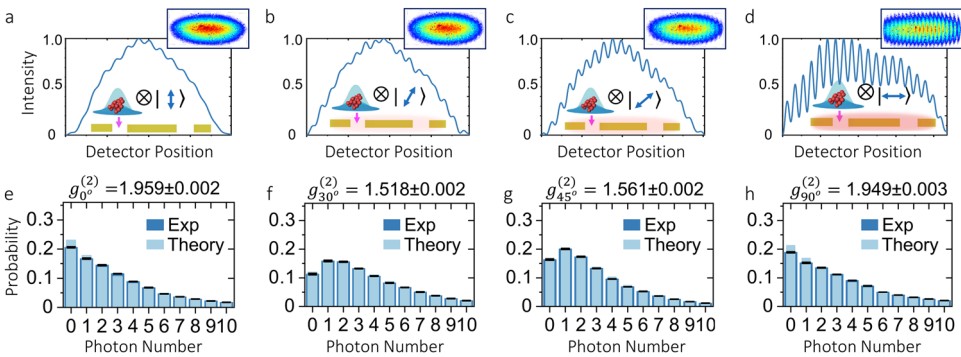

**Fig. 2 Experimental observation of plasmon-induced interference and the modification of quantum statistics.** The formation of interference fringes in the far field of the plasmonic structure with two slits is shown in panels (**a**–**d**). Panel (**a**) shows the spatial distribution of a thermal multiphoton system transmitted by a single slit. In this case, the contribution from optical near fields is negligible and no photons are transmitted through the second slit. As shown in panel (**b**), a rotation of the photon's polarization increases the presence of optical near fields in the plasmonic structure. In this case, photon-plasmon scattering processes induce small changes to the spatial distribution of the transmitted photons. Part (**c**) shows that the increasing excitation of plasmons is manifested through the increasing visibility of the interference structure. Panel (**d**), shows a clear modification of spatial coherence induced by optical near fields. Remarkably, the modification of spatial coherence induced by the presence of plasmons is also accompanied by the modification of the quantum statistical fluctuations of the field as indicated in panels (**e**–**h**). Each of these photon-number distributions corresponds to the spatial profiles above from (**a**) to (**d**). The photon-number distribution in (**e**) demonstrates that the photons transmitted by the single slit preserve their thermal statistics. Remarkably, multiparticle scattering induced by the presence of near fields modifies the photon-number distribution of the hybrid system as shown in (**f**) and (**g**). These probability distributions resemble those of coherent light sources. Interestingly, as demonstrated in (**h**), the photon-number distribution becomes thermal again when photons and plasmons are polarized along the same direction. The error bars represent the standard deviation of ten datasets. Each dataset consists of ~400,000 photon-number-resolving measurements.

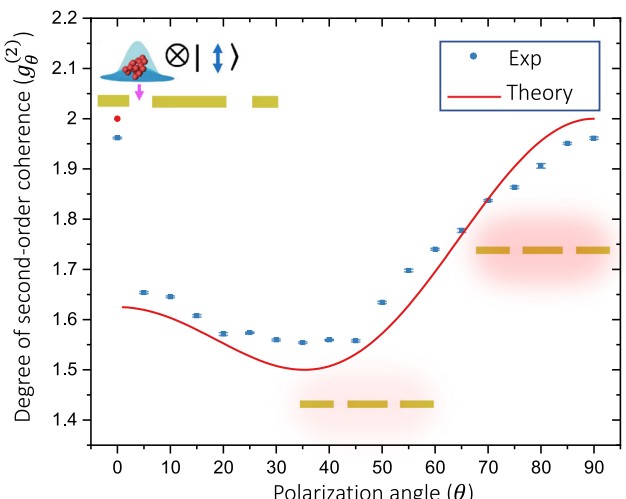

**Fig. 3 Modification of the quantum statistics of a single multiparticle system in a plasmonic structure.** The experimental data is plotted together with the theoretical prediction for the degree of the second-order correlation function. The theoretical model is based on the photon-number distribution described by Eq. (2) for a situation in which $\bar{n}_S = 3\bar{n}_{Pl}$. The error bars represent the standard deviation of ten realizations of the experiment. Each experiment consists of ~100,000 photon-number-resolving measurements.

sources, the degree of second-order coherence $g^{(2)}$ only depends on the ratio of $\bar{n}_S$ and $\bar{n}_{Pl}$ (see Supplementary Note 2). In addition, we note that the data point at $\theta = 0°$ is obtained by directly calculating $g^{(2)}$ for a single-mode thermal source. In this case, a single polarized light beam cannot be treated as two independent sources, thus one cannot use Eq. (2). Furthermore, we would like to point out that the identification of the transition point in Fig. 3, specifically the transition from one-to-two source (mode) representation is an interesting but complicated task. Indeed, there have been technical

studies that aim to develop tools to demonstrate full statistical mode reconstruction without a priori information[37]. Nevertheless, the theoretical $g^{(2)}$ after the transition is calculated using Eq. (2). The remarkable agreement between theory and experiment validates our observation of the modification of quantum statistics of plasmonic systems.

The multiparticle near-field dynamics observed for a thermal system can also induce interactions among independent multiphoton systems. We demonstrate this possibility by illuminating the plasmonic structure with two independent thermal sources. The polarization of one source is fixed to 0° whereas the polarization of the other is rotated by the angle $\theta$. This is illustrated in the central inset of Fig. 1c. In this case, both multiphoton sources are prepared to have same mean photon numbers. In Fig. 4, we report the modification of the quantum statistics of a multiphoton system comprising two modes that correspond to two independent systems. As shown in Fig. 4a, the confinement of electromagnetic near fields in our plasmonic structure modifies the value of the second-order correlation function. In this case, the shape of the second-order correlation function is defined by the symmetric contributions from the two thermal multiphoton systems. As expected, the quantum statistics of the initial thermal system with two sources remain thermal (see Fig. 4b). However, as shown in Fig. 4b–d, this becomes subthermal as the strength of the plasmonic near fields increase. The additional scattering paths induced by the presence of plasmonic near fields modify the photon-number distribution as demonstrated in Fig. 4c[27]. The strongest confinement of plasmonic near fields is achieved when the polarization of one of the sources is horizontal ($\theta = 90°$). As predicted by Eq. (2) and reported in Fig. 4d, the second-order coherence function for this particular case is $g^{(2)} = 1.5$.

The light-matter interactions explored in this experiment demonstrate the possibility of using either coherent or incoherent bosonic scattering to modify the quantum statistical fluctuations of multiparticle systems. These mechanisms are fundamentally different to the coherent interactions induced by the

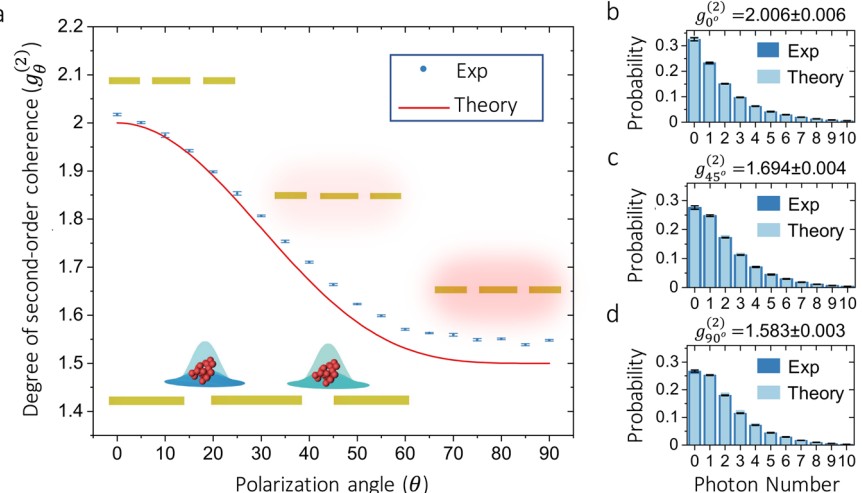

**Fig. 4 Modification of the quantum statistics of a multiphoton system comprising two sources.** The modification of the quantum statistics of a multimode plasmonic system composed of two independent multiphoton sources is shown in panel (**a**). Here, we plot experimental data together with our theoretical prediction for the degree of second-order coherence. The theoretical model is based on the photon-number distribution predicted by Eq. (2) for two independent multiphoton systems with thermal statistics and same mean photon numbers satisfying $\bar{n}_S = \bar{n}_{Pl}$. As demonstrated in (**b**), the photon-number distribution is thermal for the scattered multiphoton system in the absence of near fields ($\theta = 0°$). However, an anti-thermalization effect takes place as the strength of the plasmonic near fields is increased ($\theta = 45°$), this is indicated by the probability distribution in (**c**). Remarkably, as reported in (**d**), the degree of second-order coherence of the hybrid photonic–plasmonic system is 1.5 when plasmonic near fields are strongly confined ($\theta = 90°$). These results unveil the possibility of using plasmonic near fields to manipulate coherence and the quantum statistics of multiparticle systems. The error bars represent the standard deviation of ten realizations of the experiment. Each experiment consists of ~100,000 photon-number-resolving measurements.

indistinguishability of two bosons in Hong–Ou–Mandel interference[18,19]. Indeed, the efficient preparation of indistinguishable single surface plasmons has enabled different forms of plasmonic quantum interference[13,18–20]. In addition, the flexibility of plasmonic platforms has led to the preparation of coherent plasmonic states with tunable probability amplitudes[13,20]. Nevertheless, the probability distribution in Eq. (2) describes coherent effects, that produce interference, as well as incoherent bosonic scattering. Consequently, the plasmonic control of the quantum statistical fluctuations of a multiparticle system is achieved through both distinguishable and indistinguishable processes.

These results have important implications for multimode plasmonic systems[3,4]. Recently, there has been interest in exploiting transduction of the quantum statistical fluctuations of multimode fields for applications in imaging and quantum plasmonic networks[9,24]. Nevertheless, the modification of quantum statistics in plasmonic systems had remained unexplored[3]. Interestingly, the possibility of modifying quantum statistics and correlations through multiparticle interactions has been demonstrated in nonlinear optical systems, photonic lattices, and Bose–Einstein condensates[35,38–40]. Moreover, similar quantum dynamics have been explored for electrons in solid-state devices and cold Fermi gases[39–44]. However, our work unveils the potential of optical near fields as an additional degree of freedom to manipulate multiparticle quantum systems. This mechanism offers alternatives for the implementation of quantum control in plasmonic platforms. More specifically, our work shows that plasmonic near fields offer deterministic paths for tailoring photon statistics. In this case, the strength of plasmonic fields is deterministically controlled through polarization. Furthermore, plasmonic platforms offer practical methods for exploring controlled thermalization (and anti-thermalization) of arbitrary light fields. The modification of amplitude and phase of multiphoton systems has been explored in nonlinear optical systems[38] and photonic lattices with statistical disorder[35].

The experiments performed in plasmonic platforms in which interactions are restrained to single-particle scattering led researchers to observe the preservation of the quantum statistical properties of bosons[1–6,6–11]. These experiments triggered the idea of the conservation of quantum statistics in plasmonic systems[6,8,10,11,18,20–24]. In this article, we report the observation of the modification of the quantum statistics of multiparticle systems in plasmonic platforms. We validate our experimental observations through the theory of quantum coherence and demonstrate that multiparticle scattering mediated by dissipative near fields enables the manipulation of the excitation mode of plasmonic systems. Our findings unveil mechanisms to manipulate multiphoton quantum dynamics in plasmonic platforms. These possibilities have important implications for the fields of quantum photonics, quantum many-body systems, and quantum information science[3,16,28].

## Methods

**Sample design.** Full-wave electromagnetic simulations were conducted using a Maxwell's equation solver based on the finite difference time domain method (Lumerical FDTD). The dispersion of the materials composing the structure was taken into account by using their frequency-dependent permittivities. The permittivity of the gold film was obtained from ref. [45], the permittivity of the glass substrate (BK7) was taken from the manufacturer's specifications, and the permittivity of the index matching fluid was obtained by extrapolation from the manufacturer's specification.

As shown in Fig. 5, our nanostructure shows multiple plasmonic resonances at different wavelengths. This enables the observation of the multiphoton effects studied in this article at multiple wavelengths.

**Sample fabrication.** The sample substrates are made from SCHOTT D 263 T eco Thin Glass with a thickness of ~175 μm, polished on both sides to optical quality. The glass substrates were subsequently rinsed with acetone, methanol, isopropyl alcohol, and deionized water. The substrates were dried in nitrogen gas flow and heated in the clean oven for 15 min. Then we deposited 110-nm-thickness gold thin films directly onto the glass substrates using a Denton sputtering system with 200 W DC power, 5 mTorr argon plasma pressure, and 180 s pre-deposition conditioning time. The slit patterns were structured by Ga ion beam milling using a Quanta 3D FEG Dual beam system. The slit pattern consisted of 40-μm long slits

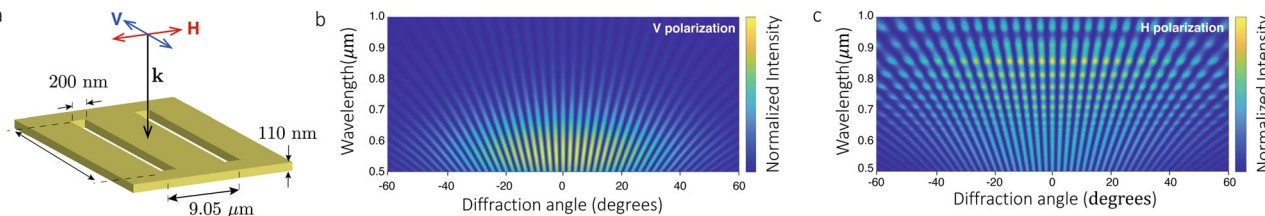

**Fig. 5 Design of plasmonic nanostructures.** The design of our plasmonic sample is shown in (**a**). The wavelength-dependent far-field interference pattern as a function of the diffraction angle is shown in (**b**). In this case, the structure is illuminated with vertically-polarized photons and no plasmonic near fields are excited. The figure in (**c**) shows a modified interference structure due to the presence of plasmonic near fields. In this case, the illuminated photons are polarized along the horizontal direction.

with a separation of 9.05 μm. While fabricating the different slit sets, each slit is machined separately. To ensure reproducibility, proper focusing of the FIB was checked by small test millings and if needed the FIB settings were readjusted accordingly to provide a consistent series of testing slits.

**Experiment.** As shown in Fig. 1c, we utilized a continuous-wave (CW) laser operating at a wavelength of 780 nm. We generated two independent sources with thermal statistics by dividing a beam with a 50:50 beam splitter. Then, we focused the beams onto two different locations of a rotating ground-glass[30]. The two beams were then coupled into single-mode fibers to extract a single transverse mode with thermal statistics (see Supplementary Note 3). In addition, we certified the thermal statistics of the photons emerging through each of the slits (see Supplementary Note 4). We attenuated the two beams with neutral-density (ND) filters to tune their mean photon numbers. The polarization state of the two thermal beams was controlled by a pair of polarizers and half-wave plates. The two prepared beams were then combined using a 50:50 beam splitter. The combined beam was weakly focused to a 450 nm spot onto the plasmonic structure that was mounted on a motorized three-axis translation stage. This enabled us to displace the sample in small increments. Once the imaging conditions were fixed, we did not modify the position of the plasmonic sample. Furthermore, we used two infinity-corrected oil-immersion microscope objectives (NA = 1.4, magnification of ×60 and working distance of 130 mm) to focus and collect light to and from the plasmonic structure. To observe the interference fringes in the far field, we built an imaging system to form the Fourier plane at ~40 cm from the plasmonic sample. Then, we characterized our plasmonic sample using horizontally polarized light. We note that the input photons can be coupled to plasmonic modes at the second slit or transmitted through the first slit. The transmission coefficient is given by the normalized transmitted intensity $I_1$ for the first slit. This was experimentally estimated as $I_1 = 0.608$. Similarly, the photon–plasmon conversion efficiency is given by the normalized transmitted intensity $I_2$ from the second slit. This coefficient was experimentally estimated as $I_2 = 0.028$. The light collected by the objective was then filtered using a 4f-imaging system to achieve specific particle number conditions for the photonic $n_s$ and plasmonic $n_{Pl}$ modes. The experiment was formalized by coupling light, using a microscope objective (NA = 0.25, magnification of ×10 and working distance of 5.6 mm), into a polarization-maintaining (PM) fiber. This fiber directs photons to a superconducting nanowire single-photon detector (SNSPD) that performs photon number resolving detection[33,34].

## Data availability

The processed experimental data generated in this study have been deposited in the figshare database at https://doi.org/10.6084/m9.figshare.15161208.

## Code availability

The code used to analyze the data and the related simulation files are available from the corresponding author upon reasonable request.

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

## Acknowledgements

C.Y., M.H., F.M. and O.S.M-L. acknowledge funding from the U.S. Department of Energy, Office of Basic Energy Sciences, Division of Materials Sciences and Engineering under Award DE-SC0021069. I.D.L. acknowledges the support of the Federico Baur Endowed Chair in Nanotechnology. R.J.L.-M. thankfully acknowledges financial support by CONACyT under the project CB-2016-01/284372 and by DGAPA-UNAM under the project UNAM-PAPIIT IN102920. We thank Dr. Dongmei Cao and the Shared Instrumentation Facility at Louisiana State University for their help in the fabrication of the plasmonic samples.

## Author contributions

The experiment was designed by C.Y., M.H., I.D.L., R.J.L.M. and O.S.M.L. The theoretical description was developed by R.J.L.M., M.A.Q.J., O.S.M.L. and C.Y. The numerical analysis was carried out by I.D.L. and F.M. The experiment was performed by M.H., C.Y., N.B. and J.F. The samples were fabricated by J.C. and J.G. The data was analyzed by C.Y. and M.H. The idea was conceived by O.S.M.L. and C.Y. The project was supervised by O.S.M.L. All authors contributed to the preparation of the manuscript.

## Competing interests

The authors declare no competing interests.
