## [Peer Review File · Nature Communications]

Reviewers' Comments:

Reviewer #1:

Remarks to the Author:

In the paper by You et al., the authors study the modification of the quantum statistics of plasmonic systems. In their work they show that multiparticle scattering effects induced by confined optical near fields can lead to a modification of the statistics of light passing through a structure made of two slits. I find the work interesting and think that it is potentially publishable. It will be useful for researchers and engineers working on developing plasmonic devices that utilize near-field quantum effects. There are a number of main concerns I have about the work and the authors would need to address these.

1. First, and perhaps the most pressing concern. The authors claim in the abstract (and elsewhere similarly) that 'For almost two decades, it has been believed that the quantum statistical properties of bosons are preserved in plasmonic systems.' This is somewhat of a generalization and it unfortunately leads to the false impression that the authors have obtained a ground-breaking result, which I believe is not the case. Previous work on the topic of the preservation of quantum statistics in plasmonic systems focused on the preservation during the transfer, or conversion process from photons to plasmons, whether the source of light was a coherent state, a single photon or some other state. In those cases, the quantum statistics and bosonic nature were observed to be well preserved, as summarized in Refs. [1-7], for example and modelled theoretically in papers cited within.

In the present work, in the conversion process of photons to plasmons, the statistics and bosonic nature are also well preserved, as expected, otherwise the authors wouldn't be able to use the Glauber-Sudarshan theory of optical coherence. The difference between the present work and previous studies seems to be that the statistics are found to vary under a more complicated interference scenario. However, all of this can be predicted from a theory where there is a complete preservation of bosonic quantum statistics during the transfer processes and then interference is considered between bosonic entities. The same scenario and modification the authors study could in principle be constructed for light on its own without any reference to plasmonics using an interferometer with some extra components, due to the linear nature of the interactions. In that case, one would not say that the quantum statistical properties of bosons are not preserved in each of the stages. Overall, they are not preserved of course, but this phenomenon isn't restricted to plasmons only. The authors need to clarify the importance of their work in relation to previous work and give a better context of their observation in the abstract, main text and summary. Finally, while the present work focuses on propagating plasmons, I look forward to seeing how the results translate to the interaction with localized plasmons, such as those in arrays of nanoparticles and metamaterials, where an analogous all-photon scenario becomes harder to find.

2. Second, the authors mention that 'we generalize our single-mode observations to a multimode system comprising two independent wavepackets'. As far as I can see, the theory and experiment do not use wavepackets at all. The theory uses a single-mode treatment (frequency), which can be a single spatial mode or multi-spatial mode, but it does not allow for the treatment of wavepackets, which would require a more advanced 'continuous-mode' frequency treatment. On the other hand, the experiment uses a CW laser (as mentioned in the supplementary information) which again is not a wavepacket. The authors would need to use a pulsed laser source in order to consider wavepackets in their experiment. This aspect should be addressed at the theory and experimental level, and any mention of wavepackets clarified when used. Perhaps the use of the word 'state' may be more appropriate instead.

3. Below equation (1), the authors should show briefly how the number distribution is extracted from the state in the coherent state basis. This is not clear.

4. More details of how equation (2) is obtained are needed.

5. For the population plots in figure 2, the value of g_2 is given also. The method/formula for extracting g_2 from the populations should be given.

Reviewer #2:

Remarks to the Author:

The paper Observation of the Modification of Quantum Statistics of Plasmonic Systems by Chenglong You and co-workers aims at demonstrating that a plasmonic device can change the quantum statistics of the exciting light. The idea is appealing but the demonstration is not convincing to me yet, with far too much unclear statements: I cannot accept this paper. I will try to argue on that.

The very simple device chosen by the author is similar to Young's slits, in which only a single slit or both of them can be illuminated by an incident light. A chaotic light source is mimicked by a laser at 780nm added to a rotating ground glass. When a single slit is illuminated, they can observe interference fringes depending on the polarization orientation. With the good polarization, surface plasmons can indeed be excited along the device and thus illuminate the second slit. Diffraction by the two slits gives rise to interferences. This really nice experiment has already been done in the literature and is completely classical. At that point, the authors are elliptical on the setup: they use a point detector and not a camera, and the light is collected through two microscope objectives, whose features are not given so that we do not know the size of the area which is observed by the detector. Where is the plane of the observed fringes? How do they scan the sample? The authors try then to measure the second order correlation. It should usually depend on a delay and a distance that do not appear anywhere, we can only guess that it is a temporal zero delay $g(2)$? For a chaotic light, the $g(2)$ is linearly dependent on the absolute value of $g(1)$, which will evolve with the state of the fringes that is observed: because we do not know where the measurement is done it is unclear if the result is surprising or not. This should be taken care of.

The most intriguing result is found in Fig. 2 (e-h) where it is seen that the statistics of the photon number change when the SP are more or less excited. Fig.2 f and g display clearly a Poisson distribution. Fig. e and h could represent a chaotic light, but we do not know once again what are the characteristics of the so-called thermal light. Because it is created using a laser and a rotating plate, the integration time T must absolutely be compared to the coherence time t_c . In fact, I do not know if the distributions e or h are truly sub-poissonian or poissonian with a mean photon number lower to one, which I guess would completely change the conclusions. The chaotic character of incident light itself has not been evaluated.

The claim made by the authors is very strong, but the demonstration is not solid enough to be accepted. With all the respect due to the authors and the hard work that has been performed in terms of experiments, I believe that the current work is at least not thorough enough to be published.

Reviewer #3:

Remarks to the Author:

Dear Editor,

In "Observation of the modification of quantum statistics of plasmonic systems", the authors present double-slit experiments with a twist. First it is shown that when illuminating only one of the slit openings, effects of emission from the second slit are seen as interference at the detector position. This is attributed to surface plasmons being generated at the illuminated slit and travelling to the second slit, where they are converted again as photons to reach the detector. Besides the intensity, also the photon-number distribution was recorded, and for skew settings of the linear polarizer of the input light, this distribution has non-thermal characteristics, in the sense that zero photons are no longer the most probable outcome. Similar distributions are as found when illuminating both slits with a thermal distribution of photons. The results are interpreted as a result of plasmons not only causing the interference pattern but also the change in photon statistics, which is claimed to be a paradigm change compared to existing literature where such changes in the statistics were assumed not to occur.

I find these interesting experiments but I am not convinced by the interpretation, which I think limits its impact.

I have several points that I would like to be discussed:

1. It is given for the experiments of Figs. 2 and 3 that $n_s = 3 n_{pl}$ (averages), so only relative photon numbers are given. However, from the central equation (2) it is clear that also the magnitudes (compared to unity) of these two photon numbers are important. What are the numbers for the averages of n_s and n_{pl} in the experiment?
2. Both the photons and the plasmons are assumed to have a thermal photon distribution, the plasmons at a lower temperature (average of n_{pl} is smaller than average of n_s). No measurement times are given but the situation is clearly a non-equilibrium situation with more than one temperature. Would repeating the experiment at half the light intensity but double measurement time give the same result? That is implicit when not giving the numbers, but why?
3. In the first experiment, $n_s = 3 n_{pl}$ (averages) while in the second experiment (Fig. 4) the two average photon numbers are assumed equal. Why is that? The plasmons are still generated at the one slot, travel to the other slit, and then get converted to photons again.
4. When blocking the second slit (the non-illuminated one), then all detected photons are combinations of horizontally and vertically polarized photons all coming from slit 1. There would be no interference fringes. The detected photon distribution that one would get would be given by Equation (2) but with the average of n_{pl} taken to zero in that formula, agree? Because I checked that for $n_s=3$ and $\eta = 0.5$ the formula still predicts "non-thermal" photon distributions in that case, with $p_{det}(1) > p_{det}(0)$, even though surface plasmons do not affect the detected photon distribution in that case. This is a counterexample to the main claim in the paper that surface plasmons cause not only the previously observed interference fringes but also the non-thermal photon statistics.
5. I suggest that as a check the authors repeat the experiment of Fig. 2 but with the light coming from the second slit blocked from going to the detector. If non-thermal photon distributions are seen, then SPPs cannot be the cause, contrary to the claims in the manuscript. If thermal photon distributions are seen, then Eq. (2) does not describe them.
6. The observed deviations from thermal photon statistics are small. There are no errors given in the experimentally measured photon distributions in Fig. 2e-f for example. The quality of the fits of the two thermal distributions in 2e and 2h is not given either.
7. From the Equation (2) it is not clear to me that energy in horizontally polarized light is converted into plasmons. I would expect that the equation would feature the photon-to-plasmon conversion efficiency.

June 17, 2021

Response to the Referee
Manuscript: NCOMMS-21-14959-T/You

Summary response statement,

Our detailed response is given below. For the aid of the editor, we first provide the reviewers' comments and then our response with the corresponding modifications.

Report of Referee #1 (in blue) followed by a detailed response to each point (in black).

Comment 1: In the paper by You et al., the authors study the modification of the quantum statistics of plasmonic systems. In their work they show that multiparticle scattering effects induced by confined optical near fields can lead to a modification of the statistics of light passing through a structure made of two slits. I find the work interesting and think that it is potentially publishable. It will be useful for researchers and engineers working on developing plasmonic devices that utilize near-field quantum effects. There are a number of main concerns I have about the work and the authors would need to address these.

Reply to Comment 1: We thank the reviewer for the positive assessment of our work. We appreciate the multiple comments, as they have allowed us to improve our manuscript.

Comment 2: First, and perhaps the most pressing concern. The authors claim in the abstract (and elsewhere similarly) that 'For almost two decades, it has been believed that the quantum statistical properties of bosons are preserved in plasmonic systems.' This is somewhat of a generalization and it unfortunately leads to the false impression that the authors have obtained a ground-breaking result, which I believe is not the case. Previous work on the topic of the preservation of quantum statistics in plasmonic systems focused on the preservation during the transfer, or conversion process from photons to plasmons, whether the source of light was a coherent state, a single photon or some other state. In those cases, the quantum statistics and bosonic nature were observed to be well preserved, as summarized in Refs. [1-7], for example and modelled theoretically in papers cited within.

In the present work, in the conversion process of photons to plasmons, the statistics and bosonic nature are also well preserved, as expected, otherwise the authors wouldn't be able to use the Glauber-Sudarshan theory of optical coherence. The difference between the present work and previous studies seems to be that the statistics are found to vary under a more complicated interference scenario. However, all of this can be predicted from a theory where there is a complete preservation of bosonic quantum statistics during the transfer processes and then interference is considered between bosonic entities. The same scenario and modification the authors study could in principle be constructed for light on its own without any reference to plasmonics using an interferometer with some extra components, due to the linear nature of the interactions. In that case, one would not say that the quantum statistical properties of bosons are not preserved in each of the stages. Overall, they are not preserved of course, but this phenomenon isn't restricted to

plasmons only. The authors need to clarify the importance of their work in relation to previous work and give a better context of their observation in the abstract, main text and summary. Finally, while the present work focuses on propagating plasmons, I look forward to seeing how the results translate to the interaction with localized plasmons, such as those in arrays of nanoparticles and metamaterials, where an analogous all-photonic scenario becomes harder to find.

Reply to Comment 2: We thank the referee for the clear analysis. In agreement with the referee, we have modified various sections of our manuscript to avoid misinterpretations. Below, we summarize the changes made to the manuscript:

1. Interactions in previous experiments vs our experiment

The new version of our manuscript emphasizes the fundamental difference between the multiparticle interactions studied in our experiment and the collective single-particle processes induced in previous experiments. We highlight the fact that previous experiments have been performed on plasmonic structures in which complex multiparticle interactions were restrained. Consequently, the conservation of photon statistics observed in previous experiments results from the occurrence of simple single-particle dynamics supported by plasmonic structures. Moreover, we stress the role played by optical near fields to define the multiparticle scattering dynamics among photons and plasmons, and how this defines the quantum statistical properties of these systems.

We also remark on the importance of photon-number-resolving detection and how other measurement schemes used in previous experiments were insensitive to the quantum statistical fluctuations of plasmonic systems.

The corresponding changes are highlighted in the new version of our manuscript, these can be found in pages 1, 4 (second column) and 6 (first column).

2. Observation of multiparticle dynamics in other quantum systems

As described in our manuscript, the multiparticle scattering among photons and plasmons defines the underlying physics behind the modification of quantum statistics in plasmonic systems. As such, it is possible to use different mechanisms to control the interactions among the particles that constitute quantum many-body systems. As pointed out by the reviewer, the modification of the quantum statistics and correlations of multiparticle systems have been demonstrated in nonlinear optical systems, photonic lattices, Bose-Einstein condensates, and solid-state systems. This is now discussed in page 6 (first column) of the main manuscript.

3. Propagating and non-propagating plasmonic fields.

Our manuscript reports on the possibility of using optical near-fields to modify the quantum statistical properties of plasmonic systems. As indicated by our theory, this process depends on the strength of the near fields and not on their propagating dynamics. Consequently, the same physical effects can be observed using localized or propagating plasmons. For simplicity, in our experiment, we used a multi-slit structure that supports propagating surface plasmons. However, we now explicitly explain the possibility of

observing modification of quantum statistics through the excitation of propagating or non-propagating near fields. This is now described in page 3 (first column) of our manuscript.

Comment 3: Second, the authors mention that ‘we generalize our single-mode observations to a multimode system comprising two independent wavepackets’. As far as I can see, the theory and experiment do not use wavepackets at all. The theory uses a single-mode treatment (frequency), which can be a single spatial mode or multi-spatial mode, but it does not allow for the treatment of wavepackets, which would require a more advanced ‘continuous-mode’ frequency treatment. On the other hand, the experiment uses a CW laser (as mentioned in the supplementary information) which again is not a wavepacket. The authors would need to use a pulsed laser source in order to consider wavepackets in their experiment. This aspect should be addressed at the theory and experimental level, and any mention of wavepackets clarified when used. Perhaps the use of the word ‘state’ may be more appropriate instead.

Reply to Comment 3: We thank the referee for providing this comment. It shows that some readers could be misled by the use of “wavepackets”. We would like to point out that this term is commonly used to describe thermal light sources with multiple photons in one mode. For example, in ref. [31] of our manuscript, Smith and Shih use this term to refer to a random field with multiple photons. We quote the following sentences from their paper:

“we have employed a standard monochromatic pseudothermal light source [24] consisting of a rotating ground glass and a single-frequency laser beam of wave-length $\lambda = 532$ nm. Millions of tiny diffusers within the rotating ground glass scatter the laser beam into many independent wave packets, or subfields, at the single-photon level with random relative phases, artificially simulating a natural thermal light source such as the sun.”

Nevertheless, for sake of clarity, we have decided to replace the word “wavepacket” with “state” or “system”. These terms do not compromise the accuracy of our theoretical description or the multiple experimental discussions in the manuscript.

Comment 4: Below equation (1), the authors should show briefly how the number distribution is extracted from the state in the coherent state basis. This is not clear.

Reply to Comment 4: We thank the referee for the comment. For any P-function $P(\alpha)$, one can obtain the corresponding photon number distribution through $p(n) = \langle n | \rho | n \rangle$, where $\rho = \int P(\alpha) |\alpha\rangle \langle \alpha| d^2\alpha$. We have now included a full discussion in the new version of the Supplementary Information. We have also provided relevant information on page 4 (first column) of our manuscript.

Comment 5: More details of how equation (2) is obtained are needed.

Reply to Comment 5: We thank the referee for the comment. We have now included the full derivation of our theory in the Sec. I of the Supplementary Information.

Comment 6: For the population plots in figure 2, the value of g_2 is given also. The method/formula for extracting g_2 from the populations should be given.

Reply to Comment 6: We thank the referee for this comment. For a single-mode field, the degree of second-order coherence is given by $g^{(2)}(\tau) = 1 + [(\langle(\Delta\hat{n})^2\rangle - \langle\hat{n}\rangle^2)]/\langle\hat{n}\rangle^2$. Here, the terms $\langle(\Delta\hat{n})^2\rangle$ and $\langle\hat{n}\rangle$ can be extracted from the measured photon-number distributions. We have now included this information in our manuscript on page 4 (second column). In addition, we refer readers to our previous work in *Appl. Phys. Rev.* 7(2), 021404 where we discuss the computation of this parameter from photon-number-resolving measurements.

Report of Referee #2 (in blue) followed by a detailed response to each point (in black).

Comment 1: The paper Observation of the Modification of Quantum Statistics of Plasmonic Systems by Chenglong You and co-workers aims at demonstrating that a plasmonic device can change the quantum statistics of the exciting light. The idea is appealing but the demonstration is not convincing to me yet, with far too much unclear statements: I cannot accept this paper. I will try to argue on that.

Reply to Comment 1: We thank the reviewer for reading our paper. We concur with the idea that additional information could make our manuscript more appealing to general audiences. For this reason, we have prepared a new version of our manuscript and additional supplementary information. We have included more experimental data, a full derivation of our theory, and additional discussions.

Comment 2: The very simple device chosen by the author is similar to Young's slits, in which only a single slit or both of them can be illuminated by an incident light. A chaotic light source is mimicked by a laser at 780nm added to a rotating ground glass. When a single slit is illuminated, they can observe interference fringes depending on the polarization orientation. With the good polarization, surface plasmons can indeed be excited along the device and thus illuminate the second slit. Diffraction by the two slits gives rise to interferences. This really nice experiment has already been done in the literature and is completely classical. At that point, the authors are elliptical on the setup: they use a point detector and not a camera, and the light is collected through two microscope objectives, whose features are not given so that we do not know the size of the area which is observed by the detector. Where is the plane of the observed fringes? How do they scan the sample? The authors try then to measure the second order correlation. It should usually depend on a delay and a distance that do not appear anywhere, we can only guess that it is a temporal zero delay $g(2)$? For a chaotic light, the $g(2)$ is linearly dependent on the absolute value of $g(1)$, which will evolve with the state of the fringes that is observed: because we do not know where the measurement is done it is unclear if the result is surprising or not. This should be taken care of.

Reply to Comment 2: We thank the reviewer for the comments. Below we address each of them.

Experimental Setup

Given the importance of the microscopes used to illuminate and collect light from the sample, we had included relevant details in the Supplementary Information of the first version of our manuscript. We have now complemented this information by reporting additional information regarding the optics used to collect light into the fiber. We also provide information related to the spot size at the sample plane. This information can be found on page 7 (first column).

As now explicitly mentioned in the caption of Fig. 2 and “Experiment” section of the Methods, the observed fringes in Fig 2(a)-(d) were recorded in the far field of the plasmonic structure with a CCD camera. The imaging system was designed to form the Fourier plane at approximately 40 cm from the plasmonic nanostructure. Moreover, we stress the fact that we do not scan the sample once the imaging conditions are fixed. However, as we indicated in the Supplementary Information of the first version of our manuscript, we mounted our sample on a motorized three-axis translation stage with 5 nm resolution. This setup allowed us to align our experimental setup and to identify the correct slit pattern. Once this step is completed, we do not modify the position of the sample, instead, we perform photon-number-resolving measurements as discussed in our manuscript.

Degree of second-order correlation function

In our experiment, we measure the second-order correlation $g^{(2)}$ to characterize and quantify the quantum statistics of our plasmonic system. As now mentioned in page 4 (second column) of the revised manuscript, the second-order correlation $g^{(2)}$ is independent of time for single-mode fields. This is a well-known result and thus we refer readers to Mandel and Wolf and the famous paper by Mandel where he used the time independence of the $g^{(2)}$ function to introduce the now so-called Mandel parameter [Opt. Lett. 4, 205 (1979)].

Regarding $g^{(2)}$ as a function of $g^{(1)}$, the formula $g^{(2)}(\tau) = 1 + |g^{(1)}(\tau)|^2$ alluded by the reviewer applies to multimode thermal fields. This can be applied for a situation in which multimode thermal light is collected by a bucket “free-space multimode” detector. However, this is not the case in our experiment. In our case, the thermal sources are single-mode and we collect light using a polarization maintaining single-mode fiber. As now described in page 4 (second column) of our manuscript, we use $g^{(2)}(\tau) = 1 + [(\langle(\Delta\hat{n})^2\rangle) - \langle\hat{n}\rangle]/\langle\hat{n}\rangle^2$ which is independent of time and valid for a single-mode field. In our experiment, we directly use the measured photon-number distributions to estimate the terms $\langle(\Delta\hat{n})^2\rangle$ and $\langle\hat{n}\rangle$, and thus the $g^{(2)}$ function. We refer readers to our previous work in *Appl. Phys. Rev.* 7(2), 021404 where we discuss the computation of this parameter from photon-number-resolving measurements.

Comment 3: The most intriguing result is found in Fig. 2 (e-h) where it is seen that the statistics of the photon number change when the SP are more or less excited. Fig.2 f and g display clearly a Poisson distribution. Fig. e and h could represent a chaotic light, but we do not know once again what are the characteristics of the so-called thermal light. Because it is created using a laser and a rotating plate, the integration time T must absolutely be compared to the coherence time t_c . In fact, I do not know if the distributions e or h are truly sub-poissonian or poissonian with a mean photon number lower to one, which I guess would completely change the conclusions. The chaotic character of incident light itself has not been evaluated.

Reply to Comment 3: We thank the reviewer for providing this comment. We agree with the reviewer that it would be insightful to provide information regarding the characterization of our light sources. Consequently, we have now included the section “Characterization of Thermal Light Sources” in the Supplementary Information. Here we show the single mode nature of our sources. For our measurements, we set the integration time of our photon counting method to 1 μ s, which corresponds to the coherence time of our CW laser. Furthermore, to demonstrate that our pseudo-thermal sources indeed possess thermal photon statistics, we measure the photon statistics of our sources. As shown in Fig. R1 below, our two thermal sources have a $g^{(2)}$ close to 2, and the experimental photon number distribution agrees well with the theoretical prediction of thermal statistics.

Figure R1: Histograms displaying theoretical and experimental photon number probability distributions for our pseudo-thermal light sources. The calculated second-order correlation function $g^{(2)}$ certifies the properties of our sources.

Consequently, the modification of the quantum statistics reported in Fig. 2 of our manuscript is indeed induced by optical near-fields as discussed therein.

Regarding the Poissonian and super-Poissonian nature of the distributions shown in Fig. 2, the mean photon numbers are 3.58 and 4.07 for the distributions shown in Fig. 2e and h. As reported in these panels, the calculated values for the $g^{(2)}$ function certify the thermal properties of these photon-number distributions.

Comment 4: The claim made by the authors is very strong, but the demonstration is not solid enough to be accepted. With all the respect due to the authors and the hard work that has been performed in terms of experiments, I believe that the current work is at least not thorough enough to be published.

Reply to Comment 4: We sincerely thank the reviewer for all the comments. We believe that the new discussions and additional experimental data in our manuscript will resolve any possible loopholes identified by the reviewer.

Report of Referee #3 (in blue) followed by a detailed response to each point (in black).

Comment 1: In "Observation of the modification of quantum statistics of plasmonic systems", the authors present double-slit experiments with a twist. First it is shown that when illuminating only one of the slit openings, effects of emission from the second slit are seen as interference at the detector position. This is attributed to surface plasmons being generated at the illuminated slit and travelling to the second slit, where they are converted again as photons to reach the detector. Besides the intensity, also the photon-number distribution was recorded, and for skew settings of the linear polarizer of the input light, this distribution has non-thermal characteristics, in the sense that zero photons are no longer the most probable outcome. Similar distributions are as found when illuminating both slits with a thermal distribution of photons. The results are interpreted as a result of plasmons not only causing the interference pattern but also the change in photon statistics, which is claimed to be a paradigm change compared to existing literature where such changes in the statistics were assumed not to occur.

I find these interesting experiments but I am not convinced by the interpretation, which I think limits its impact.

Reply to Comment 1: We thank the reviewer for the positive assessment of our work. Below, we address the concerns raised by the reviewer.

Comment 2: I have several points that I would like to be discussed:

1. It is given for the experiments of Figs. 2 and 3 that $n_s = 3 n_{pl}$ (averages), so only relative photon numbers are given. However, from the central equation (2) it is clear that also the magnitudes (compared to unity) of these two photon numbers are important. What are the numbers for the averages of n_s and n_{pl} in the experiment?

Reply to Comment 2: We thank the reviewer for the comment. We agree that the probability function in Eq. (2) depends on the magnitude of n_s and n_{pl} . However, this is not an absolute metric to quantify the changes in the quantum statistical properties of a physical system. As described below, the modification of the quantum statistics is quantified through the $g^{(2)}$ function, which is independent of the mean photon numbers. Indeed, the $g^{(2)}$ function depends only on the ratio between n_s and n_{pl} . We have collected additional experimental data to support this point. The reviewer can find the corresponding discussion in Section II of the Supplementary Information. As shown below, we have performed the experiments discussed in Fig. 2f and g for different mean photon numbers. In this case, the $g^{(2)}$ function remains unchanged, however, the probability distribution described by Eq. (2) predicts different photon-number distributions.

Figure R2: Intensity independence of the modification of quantum statistics for our plasmonic system. Panels **a** to **d** show the probability distribution and the value of the $g^{(2)}$ function for a situation in which the sample is illuminated with linearly polarized light at 30° . While the photon number distribution changes with the brightness of the source, the value of the $g^{(2)}$ remains unchanged. This behavior shows the relevance of the $g^{(2)}$ function as a metric to quantify the quantum statistical fluctuations of a physical system. For sake of completeness, panels **e** to **h** show a similar trend for the case in which the sample is illuminated with diagonally polarized light. These probability distributions demonstrate that the second-order quantum coherence function $g^{(2)}$ does not change with respect to the brightness of the experiment.

For sake of clarity, in page 4 (second column) of the main paper, we now refer readers to Section II of the Supplementary Information.

Comment 3: Both the photons and the plasmons are assumed to have a thermal photon distribution, the plasmons at a lower temperature (average of n_{pl} is smaller than average of n_s). No measurement times are given but the situation is clearly a non-equilibrium situation with more than one temperature. Would repeating the experiment at half the light intensity but double measurement time give the same result? That is implicit when not giving the numbers, but why?

Reply to Comment 3: We thank the reviewer for the comment. We would like to point out that while our sources can be directly related to blackbody radiation, our experiment uses pseudo-thermal light sources such as those used by Arecchi in Phys. Rev. Lett. 19, 1168 (1967) or Smith and Shih in Phys. Rev. Lett. 120, 063606 (2018). As such, our experiment does not depend on the physical temperature surrounding our plasmonic sample. In fact, our experiment was performed at room temperature. Therefore, the total measurement time does not play a critical role in the measured photon statistics. For sake of completeness, below we show the independence of the photon statistics on different integration times.

Figure R3: Histograms displaying theoretical and experimental photon number probability distributions of the output for different measurement times and intensities. The total measurement time is set to 1s, 2s and 4s for the panel **a**, **b** and **c**, respectively. We also adjust the intensity of the input light so the total number of events recorded in the experiment remains the same.

Comment 4: In the first experiment, $n_s = 3 n_{pl}$ (averages) while in the second experiment (Fig. 4) the two average photon numbers are assumed equal. Why is that? The plasmons are still generated at the one slot, travel to the other slit, and then get converted to photons again.

Reply to Comment 4: We thank the reviewer for this question. As discussed in “Reply to Comment 2”, the modification of the quantum statistics and the $g^{(2)}$ function depend on the ratio between n_s and n_{pl} . Consequently, we decided to show different multiparticle interactions by using a different ratio between n_s and n_{pl} . These two cases illustrate different behaviors for the $g^{(2)}$ function and consequently for the modification of quantum statistics for plasmonic systems.

Comment 5: When blocking the second slit (the non-illuminated one), then all detected photons are combinations of horizontally and vertically polarized photons all coming from slit 1. There would be no interference fringes. The detected photon distribution that one would get would be given by Equation (2) but with the average of n_{pl} taken to zero in that formula, agree? Because I checked that for $n_s=3$ and $\eta = 0.5$ the formula still predicts “non-thermal” photon distributions in that case, with $p_{det}(1) > p_{det}(0)$, even though surface plasmons do not affect the detected photon distribution in that case. This is a counterexample to the main claim in the paper that surface plasmons cause not only the previously observed interference fringes but also the non-thermal photon statistics.

Reply to Comment 5: We thank the reviewer for this comment. Indeed, the reviewer’s calculation is correct; however, the condition $n_{pl} = 0$, $n_s = 3$ and $\eta = 0.5$ does not fall within the validity of Eq. (1) and (2). As originally stated by Glauber (see Ref. [13]), the superposition or convolution law [Eq. (1)] can be used only when assuming that there are two different radiation sources, whether they are distinguishable or indistinguishable, coupled to the field measured at the detector. When taking $n_{pl} = 0$, $n_s = 3$, and $\eta = 0.5$, one is assuming that photons at the detector come only from one source, i.e. slit 1, and thus cannot be thought of as a combination of horizontally and vertically polarized fields. In this way, by blocking (or turning off) one of the thermal sources, one would expect to detect a thermal photon distribution, with a $g^{(2)}$ ideally equal to 2. Please note that we have verified this experimentally, see “Reply to Comment 6” for details.

To clarify this point, we have included new text on pages 3 (second column) and 4 (first column) of the revised manuscript as well as the Sec I of the Supplementary Information, to clearly state

that Eq. (1) and (2) are valid only when the two sources, that is, slit 1 and slit 2 are active and contribute to the resulting combined field at the detector.

Comment 6: I suggest that as a check the authors repeat the experiment of Fig. 2 but with the light coming from the second slit blocked from going to the detector. If non-thermal photon distributions are seen, then SPPs cannot be the cause, contrary to the claims in the manuscript. If thermal photon distributions are seen, then Eq. (2) does not describe them.

Reply to Comment 6: We thank the reviewer for the suggestion. As shown in Fig. R4, when blocking either of the slits, we observe a thermal photon distribution. Indeed, as we argued above, Eqs. (1) and (2) do not describe independent single-source distributions. Instead, these equations describe the photon distribution when two independent, distinguishable, or indistinguishable thermal fields are combined.

Figure R4: Histograms displaying theoretical and experimental photon number probability distributions of the output from either slit. In **a**, the second slit, where the SPPs are transmitted, is blocked. Therefore, the probability distribution corresponds to the transmitted thermal beam. In **b**, the first slit is blocked, thus the probability distribution represents the quantum statistics of the plasmonic mode.

We have now included this information in Sec. IV of the Supplementary Information.

Comment 7: The observed deviations from thermal photon statistics are small. There are no errors given in the experimentally measured photon distributions in Fig. 2e-f for example. The quality of the fits of the two thermal distributions in 2e and 2h is not given either.

Reply to Comment 7: We thank the reviewer for the suggestion. We have re-plotted Fig. 2 and Fig. 4 with error bars. In addition, we note that we do not fit these measured photon number distributions. Instead, the theoretical photon number distributions in Fig. 2 are defined by Eq. (2) using the measured mean photon number. Furthermore, these calculated values for the $g^{(2)}$ function in Figs. 2e and **h** certify the thermal properties of these photon-number distributions.

Comment 8: From the Equation (2) it is not clear to me that energy in horizontally polarized light is converted into plasmons. I would expect that the equation would feature the photon-to-plasmon conversion efficiency.

Reply to Comment 8: We thank the reviewer for the question. We performed a full characterization of our sample. We estimate that the transmission coefficient of the first slit is around 60.8%. Moreover, the photon-to-plasmon conversion efficiency, represented by the output from the second slit, is approximately 2.8% of the input beam. These numbers were obtained using FDTD simulations and were experimentally confirmed. We have included this information in the “Experiment” section of the Methods in our manuscript.

We did not include the efficiencies in Eq. (2) as we tried to keep a general and simple description of the complex interactions in our plasmonic sample. However, we note that Eq. (2) can be written in a different form to incorporate the photon-to-plasmon conversion efficiency. To achieve this, one can simply replace \bar{n}_s to $\eta_1 \bar{n}$ and \bar{n}_{pl} to $\eta_2 \eta \bar{n}$, respectively. Here η_1 and η_2 represent the overall coupling efficiency of the photonic and plasmonic modes, respectively, \bar{n} represents the input light intensity, and η has the same definition as in Eq. (2).

Reviewers' Comments:

Reviewer #1:

Remarks to the Author:

I read the reply letter from the authors and their revised manuscript. They have done a good job to address most of my concerns and it appears those of the other reviewers.

However, my first and most pressing concern has not been addressed appropriately in the revision, which includes the authors' opening statement in the abstract 'For almost two decades, it has been believed that the quantum statistical properties of bosons are preserved in plasmonic systems.' As I mentioned in my previous report, this is a generalization and it leads to the false impression that the authors have obtained a ground-breaking result, which is not the case.

The opening statement in the abstract implies that researchers working on quantum effects of plasmonic systems over the past two decades have blindly assumed that the quantum statistical properties of bosons are always preserved in plasmonic systems in all cases. As I elaborated in my report, this is not the case based on work published in the literature. Furthermore, the modification of quantum statistics that the authors observe can be easily understood from a basic understanding of quantum interference. For instance, it is not clear to what extent the authors' experiment is different to say a plasmonic Hong-Ou-Mandel experiment, which has been demonstrated many times already. Here, there is a 'multiparticle interaction' (involving 2 plasmons) where the quantum statistics in the individual output modes are clearly modified by plasmonic interference.

To better understand the above-mentioned modification of boson statistics in the context of two input spatial modes and a single output spatial mode, as in the authors' experiment (e.g. Figure 4), one only needs to look at the plasmonic Hong-Ou-Mandel experiment of Cai et al., *Physical Review Applied* 2, 014004 (2014), where the statistics (populations) of the single-mode output bosonic state (either $|0\rangle$ or $|2\rangle$) are different to those of the individual single-mode input bosonic states ($|1\rangle$). The modification of the statistics is caused by a plasmonic structure, in this case a waveguide, acting as a mediator and relies on interference, similar to the authors' study.

The modification of the quantum statistics via a plasmonic mode has also been observed for single particles on their own, for instance in the work of Kolesov et al., *Nature Physics* 5, 470 (2009), where a plasmonic Mach-Zehnder interferometer maps a single input state ($|1\rangle$) from a nitrogen vacancy center to an output state with different statistics/populations ($|0\rangle$ or $|1\rangle$).

It appears that the main difference between the authors' study and the previous one and two particle experiments mentioned above is the observation of the modification of the quantum statistics of more than 2 photons, i.e. multiphoton input states.

Furthermore, the interference effect the authors study for plasmons has been studied in many other systems, as the authors themselves acknowledge in their reply to my concern, where they write 'the modification of the quantum statistics and correlations of multiparticle systems have been demonstrated in nonlinear optical systems, photonic lattices, Bose-Einstein condensates, and solid-state systems'. The authors have even added this information to p6 of their revision and the references they give are to works spanning the last two decades.

In light of all of the above, I find the authors' opening sentence in the abstract as disingenuous and misleading to the quantum optics and plasmonics communities, and it should be changed. I would suggest something along the lines of: 'For almost two decades, researchers have observed the preservation of the quantum statistical properties of bosons in many different types of plasmonic systems.' There is a similar sentence in the summary that should be changed. I would also need to see the authors comment on the similarities and differences to the previous one and two particle plasmonic modification of quantum statistics works mentioned above.

Reviewer #2:

Remarks to the Author:

I thank the authors for their efforts to make the manuscript more clear and readable. The information about the experimental setup as well as the basics of the calculation make the paper much more interesting. I would now recommend its publication.

Reviewer #3:

Remarks to the Author:

Dear Editor,

I have read the two other referee reports, the authors' replies to all three reports, and the revised manuscript. As a general comment, I noticed that the authors have responded quickly and extensively with many careful and detailed answers and with new data. Let me now comment on the authors' replies to the points that I raised.

On Reply to Comment 1: OK.

On Reply to Comment 2: It was very helpful to see that in their experiments, the second-order correlation function does not depend on the brightness of their sources, whereas the number distribution does. So Figure 1 of the Supplement is a welcome addition.

On Reply to Comment 3: The authors show in their Figure R3 that doing the experiment at half the intensity but measuring twice as long, changes the photon distribution but not the g_2 functions. This answers my question fully.

On Reply to Comment 4: My comment was borne out of surprise that the ratio between n_s and n_{pl} could be varied, while I thought this number would be fixed. But if it can be varied, then I understand the choice that results from different fractions n_{pl} / n_s are shown.

On Reply to Comment 5: Here I am not convinced by the authors' reply. The authors argue that Eq. (1) is not valid in case one blocks the output of one of the ports. However, suppose only slit 1 is illuminated and that no photons are detected that are emitted by the second slit, either because they were not emitted or because they were blocked, or because we replaced the plasmonic material by another material that does not support plasmons. As far as photon detection is concerned, these cases can be described by a zero-temperature thermal state of the second slit, which emits no photons, and for which the P-function is a delta function. Inserting $P(\alpha') = \delta(\alpha')$ in Eq. (1) simply gives the expected result that the detected photon distribution is identical to the thermal photon distribution emitted by slit 1, a result that also Roy Glauber would agree with. Therefore, I do not agree that Eq. (1) is not valid when no photons are detected from slit 2. So the limiting situation $n_{pl} = 0$ (or $n_{pl} \ll 1$) that I brought up previously and which corresponds to (almost) no photons being emitted from slit 2 is well described by Eq. (1) and I have found no reason why Eq. (2) would then not be valid as well for $n_{pl}=0$ (or $n_{pl} \ll 1$). In that limit, the detected light is a combination of horizontal and vertical photons of only slit 1. And for that case, in combination with $n_s = 3$ and $\eta = 0.5$, Eq. (2) predicts a NON-thermal photon distribution, as I argued before. So Eq. (2) implies that merely the convolution of distributions of the two distinguishable types of photons (horizontal and vertical) coming from slit 1 can already lead to nonthermal photon distributions at the detector. Which in itself is very interesting.

In their reply the authors write that in the case of $n_{pl}=0$ "one is assuming that photons at the detector come only from one source, i.e. slit 1, and thus cannot be thought of as a combination of horizontally and vertically polarized fields." However, this contradicts the authors' own explanation of the three sources in their experiments (I cite beginning of Supplement): "The first two contributions correspond to the horizontally- and vertically-polarized fields that traverse the illuminated slit." So both horizontally and vertically polarized fields ARE emitted from slit 1 after all.

It is evident that Eq. (2) can describe non-thermal distributions $P_{det}(n)$ for $n_{pl} = 0$ or almost 0, the case where plasmons have no or a negligible effect so that the second slit emits no (or a negligible amount of) photons. The authors have not provided a quantitative limit of validity of

their Eqs. (1) or (2) that explains why Eq. (2) should not be trusted when n_{pl} becomes smaller than a certain limiting value. My quick check that $n_{pl} = 0$ gives the expected detected photon distribution in Eq. (1) makes me conclude that there is no such lower limit for Eqs. (1) and (2). So I maintain that the authors have produced interesting measurements of nonthermal photon distributions, but I am still not convinced by their interpretation that the difference between the thermal and non-thermal distributions are due to plasmons.

On Reply to Comment 6: in figure R4 and in the Supplement the authors have provided measurements of the photon distributions obtained by blocking slit 2 (panel a) and slit 1 (panel b), which is appreciated. In the provided measurements, both photon distributions come out as thermal (or close to thermal), at least distributions where $P_{det}(0) > P_{det}(1)$. Interestingly, and this was not clear to me when writing my first report, $P_{det}(0) > P_{det}(1)$ can still be in agreement with their Eq. (2) with $n_{pl}=0$ even though the authors state otherwise: for $\eta = 0.5$ and $n_{pl} = 0$, it depends on the value of n_s whether the distribution $P_{det}(n)$ of Eq. (2) is thermal-like or not: for $n_s = 1.0$, one immediately finds from Eq. (2) that $P_{det}(0) > P_{det}(1)$ so thermal-like behavior (and this value $n_s=1$ seems to correspond to Figure R4(a)), while for $n_s = 3.0$ (the case I discussed before in my first report) one finds $P_{det}(0) < P_{det}(1)$: non-thermal like behavior. In both these cases, no photons are emitted from slit 2, so the difference between these thermal and non-thermal photon distributions described by Eq. (2) cannot be explained by plasmons being present or not.

So the provided experiments in figure R4 do not prove that Eq. (2) is not valid when blocking slit 2. And in a parameter regime where theoretically it seems well valid, Eq. (2) describes non-thermal photon distributions also in the limit of negligible plasmons ($n_{pl} \ll 1$ or even $\rightarrow 0$), contrary to the authors' central claim that non-thermal photon distributions are caused by plasmons.

On Reply to Comment 7: The authors have provide error bars and this has improved the manuscript considerably.

On Reply to (the final) Comment 8: It is useful that the authors provided the transmission coefficient of the first slit and the photon-to-plasmon efficiency in the second slit. And I agree with the authors' response regarding the photon-to-plasmon conversion efficiency: one could express Eq. (2) in terms of this efficiency but one can also leave it out as the authors did. This was enlightening.

In Summary: All in all, I am quite satisfied with most of the authors' responses, but as is clear from Points 5 and 6 above, I still find myself disagreeing with their central claim that the non-thermal photon distributions are due to plasmons (which then "will establish new paradigms in quantum plasmonics"). But because of this disagreement with the central claim, I cannot recommend to accept this work for publication in its present form. My reasoning can be summarized as follows: I did a simple check that Eq. (1) gives the physically intuitive photon distributions also for very small (and even zero) average plasmon numbers, contrary to the statement by the authors that Eq. (1) stops being valid in that limit. In that limit of very small n_{pl} , plasmon effects are negligible but Eq. (2) still predicts non-thermal photon distributions $P_{det}(n)$, at least when choosing n_s large enough (and $n_s = 3$ is large enough). So that tells me that it is too quick to ascribe the non-thermal distributions as observed by the authors to plasmons. How could I be convinced by the authors? It would start here: it seems that the authors would like to use Eq. (2) for some values of n_{pl} but not for others, so I am asking them to provide a quantitative justification of the range of validity of their formula (2), i.e. for what values of n_{pl} it can be used and for what values it cannot. Such a justification at present is lacking.

Response to the Referee
Manuscript: NCOMMS-21-14959-A/You

Summary response statement,

Our detailed response is given below. For the aid of the editor, we first provide the reviewers' comments and then our response with the corresponding modifications.

Report of Referee #1 (in blue) followed by a detailed response to each point (in black).

Comment 1: I read the reply letter from the authors and their revised manuscript. They have done a good job to address most of my concerns and it appears those of the other reviewers.

However, my first and most pressing concern has not been addressed appropriately in the revision, which includes the authors' opening statement in the abstract 'For almost two decades, it has been believed that the quantum statistical properties of bosons are preserved in plasmonic systems.' As I mentioned in my previous report, this is a generalization and it leads to the false impression that the authors have obtained a ground-breaking result, which is not the case.

The opening statement in the abstract implies that researchers working on quantum effects of plasmonic systems over the past two decades have blindly assumed that the quantum statistical properties of bosons are always preserved in plasmonic systems in all cases. As I elaborated in my report, this is not the case based on work published in the literature. Furthermore, the modification of quantum statistics that the authors observe can be easily understood from a basic understanding of quantum interference. For instance, it is not clear to what extent the authors' experiment is different to say a plasmonic Hong-Ou-Mandel experiment, which has been demonstrated many times already. Here, there is a 'multiparticle interaction' (involving 2 plasmons) where the quantum statistics in the individual output modes are clearly modified by plasmonic interference.

To better understand the above-mentioned modification of boson statistics in the context of two input spatial modes and a single output spatial mode, as in the authors' experiment (e.g. Figure 4), one only needs to look at the plasmonic Hong-Ou-Mandel experiment of Cai et al., Physical Review Applied 2, 014004 (2014), where the statistics (populations) of the single-mode output bosonic state (either $|0\rangle$ or $|2\rangle$) are different to those of the individual single-mode input bosonic states ($|1\rangle$). The modification of the statistics is caused by a plasmonic structure, in this case a waveguide, acting as a mediator and relies on interference, similar to the authors' study.

The modification of the quantum statistics via a plasmonic mode has also been observed for single particles on their own, for instance in the work of Kolesov et al., Nature Physics 5, 470 (2009), where a plasmonic Mach-Zehnder interferometer maps a single input state ($|1\rangle$) from a nitrogen vacancy center to an output state with different statistics/populations ($|0\rangle$ or $|1\rangle$).

It appears that the main difference between the authors' study and the previous one and two particle experiments mentioned above is the observation of the modification of the quantum statistics of more than 2 photons, i.e. multiphoton input states.

Reply to Comment 1: We are glad that most of the reviewer's concerns were addressed in the previous revision. We also thank the reviewer for the additional specific comments. In agreement with the reviewer, we have modified our manuscript accordingly.

In regard to the misleading statements, we have used the rephrased statement provided by the reviewer (see reviewer's Comment 2) to modify the abstract. We have also modified the conclusions to reflect the suggestions provided by the reviewer. In addition, we now stress the multiphoton nature of our protocol as advised by the reviewer.

Regarding the difference between our work and the two-photon dynamics observed in previous plasmonic Hong-Ou-Mandel (HOM) experiments, we have added a new paragraph discussing the underlying physics that leads to fundamentally different mechanisms between the two processes. This discussion can be found on page 6 of our revised manuscript. We now point out that in addition to the number of interacting particles, the fundamental difference between our work and previous HOM experiments resides in the nature of the interactions. The HOM interference effect is a coherent process that results from the indistinguishability between two bosons. In contrast, our experiment results from both distinguishable and indistinguishable bosonic scattering. As described by Eq. 2, the modification of the quantum statistics of a multiparticle system in our experiment results from coherent and incoherent bosonic interactions. In this discussion, we have incorporated the relevant references provided by the reviewer.

Comment 2: Furthermore, the interference effect the authors study for plasmons has been studied in many other systems, as the authors themselves acknowledge in their reply to my concern, where they write 'the modification of the quantum statistics and correlations of multiparticle systems have been demonstrated in nonlinear optical systems, photonic lattices, Bose-Einstein condensates, and solid-state systems'. The authors have even added this information to p6 of their revision and the references they give are to works spanning the last two decades.

In light of all of the above, I find the authors' opening sentence in the abstract as disingenuous and misleading to the quantum optics and plasmonics communities, and it should be changed. I would suggest something along the lines of: 'For almost two decades, researchers have observed the preservation of the quantum statistical properties of bosons in many different types of plasmonic systems.' There is a similar sentence in the summary that should be changed. I would also need to see the authors comment on the similarities and differences to the previous one and two particle plasmonic modification of quantum statistics works mentioned above.

Reply to Comment 2: We thank the reviewer for the multiple comments. We have used the suggested sentence in the new version of our manuscript. These changes can be found in page 1 and 6 of our revised manuscript.

Report of Referee #2 (in blue) followed by a detailed response to each point (in black).

Comment 1: I thank the authors for their efforts to make the manuscript more clear and readable. The information about the experimental setup as well as the basics of the calculation make the paper much more interesting. I would now recommend its publication.

Reply to Comment 1: We thank the reviewer for the previous comments and the positive opinion on our revised manuscript.

Report of Referee #3 (in blue) followed by a detailed response to each point (in black).

Comment 1: I have read the two other referee reports, the authors' replies to all three reports, and the revised manuscript. As a general comment, I noticed that the authors have responded quickly and extensively with many careful and detailed answers and with new data. Let me now comment on the authors' replies to the points that I raised.

On Reply to Comment 1: OK.

On Reply to Comment 2: It was very helpful to see that in their experiments, the second-order correlation function does not depend on the brightness of their sources, whereas the number distribution does. So Figure 1 of the Supplement is a welcome addition.

On Reply to Comment 3: The authors show in their Figure R3 that doing the experiment at half the intensity but measuring twice as long, changes the photon distribution but not the g_2 functions. This answers my question fully.

On Reply to Comment 4: My comment was borne out of surprise that the ratio between n_s and n_{pl} could be varied, while I thought this number would be fixed. But if it can be varied, then I understand the choice that results from different fractions n_{pl} / n_s are shown.

Reply to Comment 1: We thank the reviewer for the positive feedback on the revision of our manuscript.

Comment 2: On Reply to Comment 5: Here I am not convinced by the authors' reply. The authors argue that Eq. (1) is not valid in case one blocks the output of one of the ports. However, suppose only slit 1 is illuminated and that no photons are detected that are emitted by the second slit, either because they were not emitted or because they were blocked, or because we replaced the plasmonic material by another material that does not support plasmons. As far as photon detection is concerned, these cases can be described by a zero-temperature thermal state of the second slit, which emits no photons, and for which the P-function is a delta function. Inserting $P(\alpha') =$

$\delta(\alpha')$ in Eq. (1) simply gives the expected result that the detected photon distribution is identical to the thermal photon distribution emitted by slit 1, a result that also Roy Glauber would agree with. Therefore, I do not agree that Eq. (1) is not valid when no photons are detected from slit 2. So the limiting situation $n_{pl} = 0$ (or $n_{pl} \ll 1$) that I brought up previously and which corresponds to (almost) no photons being emitted from slit 2 is well described by Eq. (1) and I have found no reason why Eq. (2) would then not be valid as well for $n_{pl}=0$ (or $n_{pl} \ll 1$). In that limit, the detected light is a combination of horizontal and vertical photons of only slit 1. And for that case, in combination with $n_s = 3$ and $\eta = 0.5$, Eq. (2) predicts a NON-thermal photon distribution, as I argued before. So Eq. (2) implies that merely the convolution of distributions of the two distinguishable types of photons (horizontal and vertical) coming from slit 1 can already lead to nonthermal photon distributions at the detector. Which in itself is very interesting.

In their reply the authors write that in the case of $n_{pl}=0$ “one is assuming that photons at the detector come only from one source, i.e. slit 1, and thus cannot be thought of as a combination of horizontally and vertically polarized fields.” However, this contradicts the authors’ own explanation of the three sources in their experiments (I cite beginning of Supplement): “The first two contributions correspond to the horizontally- and vertically-polarized fields that traverse the illuminated slit.” So both horizontally and vertically polarized fields ARE emitted from slit 1 after all.

It is evident that Eq. (2) can describe non-thermal distributions $P_{det}(n)$ for $n_{pl} = 0$ or almost 0, the case where plasmons have no or a negligible effect so that the second slit emits no (or a negligible amount of) photons. The authors have not provided a quantitative limit of validity of their Eqs. (1) or (2) that explains why Eq. (2) should not be trusted when n_{pl} becomes smaller than a certain limiting value. My quick check that $n_{pl} = 0$ gives the expected detected photon distribution in Eq. (1) makes me conclude that there is no such lower limit for Eqs. (1) and (2). So I maintain that the authors have produced interesting measurements of nonthermal photon distributions, but I am still not convinced by their interpretation that the difference between the thermal and non-thermal distributions are due to plasmons.

Reply to Comment 2: We thank the reviewer for the detailed comment. The additional information provided by the reviewer has enabled us to identify the origin of the misinterpretation of Eqs. (1) and (2). Below, we clarify these points.

Validity of Eq. (1):

We agree with the reviewer’s calculation on using a delta function for the combined P-function describing the vacuum and an arbitrary state. However, as verified by the reviewer, this calculation shows that the resulting P-function of the combined system remains the same as the original P-function of the individual system without the vacuum. In this case, the vacuum state corresponds to the description of a source that has been turned off (see, Ref. 14). In our manuscript, we referred to Eq. (1) as being valid for two active sources because no convolution operation is needed when one of them is turned off (or in the vacuum state). However, to avoid misunderstandings, we do not refer to Eq. (1) anymore on page 4 of the main text and in Section I of the Supplementary Information.

Validity of Eq. (2):

As stated in our previous response and in the revised manuscript, we reiterate that Eq. (2) is only valid when the two sources are active and contribute to the combined field measured by the detector. However, the situation considered by the reviewer in which $n_{p1} \approx 0$, $n_s = 3$ and $\eta = 0.5$ corresponds to the emission of a single slit. This is a single source scenario and Eq. (2) cannot be used to describe the quantum statistics. In this case, the horizontal and vertical components are in the same spatial mode (slit) and one cannot apply the convolution of probabilities that leads to Eq. (2). This one-source situation can be simply described by the probability distribution of a single thermal beam.

Indeed, we experimentally measured the condition considered by the reviewer. This corresponds to the first point in Fig. 3a. The statistics of this data point correspond to that of a single one-mode thermal source described by $P(n) = \bar{n}^n / (\bar{n} + 1)^{n+1}$. However, as the second source becomes active, the multiparticle scattering processes mediated by plasmonic near fields are captured and accurately described by Eq. (2). In order to clarify this point, we now discuss the limits of Eq. (2) in the context of Fig. 3 (see page 5 of our manuscript). Our theory and experiment show that in the limit of $n_{p1} = 0$, one cannot describe the statistics of the photons emitted by a single slit as the statistical combination of two *independent* polarized fields. Furthermore, this point is also experimentally verified by new data shown in Fig. R1, here we block the second slit for different polarization angles (different η). It is clear that the quantum statistics remain thermal. The reviewer can find additional discussion in Sec. IV of the Supplementary Information.

Figure R1: Histograms displaying theoretical and experimental photon-number probability distributions for the output from the first slit. The probability distribution corresponds to the transmitted thermal beam, and it is independent from the beam's polarization.

Finally, we would like to point out that the identification of the transition point in Fig. 3, specifically the transition from one-to-two source (mode) representation is an interesting but complicated task. Indeed, there have been technical studies that aim to develop tools to demonstrate full statistical mode reconstruction without *a priori* information. This has been explored by I. A. Burenkov and co-workers in Physical Review A 95(5), 053806 (2017). In order to find the transition point in Fig. 3, one would need to apply a modified version of Burenkov's approach to describe the quantum statistics induced by a mode transition. Nevertheless, this interesting point falls out of the scope of our work.

Comment 3: On Reply to Comment 6: in figure R4 and in the Supplement the authors have provided measurements of the photon distributions obtained by blocking slit 2 (panel a) and slit 1 (panel b), which is appreciated. In the provided measurements, both photon distributions come out as thermal (or close to thermal), at least distributions where $P_{\text{det}}(0) > P_{\text{det}}(1)$. Interestingly, and this was not clear to me when writing my first report, $P_{\text{det}}(0) > P_{\text{det}}(1)$ can still be in agreement with their Eq. (2) with $n_{\text{pl}}=0$ even though the authors state otherwise: for $\eta=0.5$ and $n_{\text{pl}}=0$, it depends on the value of n_{s} whether the distribution $P_{\text{det}}(n)$ of Eq. (2) is thermal-like or not: for $n_{\text{s}}=1.0$, one immediately finds from Eq. (2) that $P_{\text{det}}(0) > P_{\text{det}}(1)$ so thermal-like behavior (and this value $n_{\text{s}}=1$ seems to correspond to Figure R4(a)), while for $n_{\text{s}}=3.0$ (the case I discussed before in my first report) one finds $P_{\text{det}}(0) < P_{\text{det}}(1)$: non-thermal like behavior. In both these cases, no photons are emitted from slit 2, so the difference between these thermal and non-thermal photon distributions described by Eq. (2) cannot be explained by plasmons being present or not.

So the provided experiments in figure R4 do not prove that Eq. (2) is not valid when blocking slit 2. And in a parameter regime where theoretically it seems well valid, Eq. (2) describes non-thermal photon distributions also in the limit of negligible plasmons ($n_{\text{pl}} \ll 1$ or even $\rightarrow 0$), contrary to the authors' central claim that non-thermal photon distributions are caused by plasmons.

Reply to Comment 3: We thank the reviewer for this question. Following our reply in the previous point, we would like to mention again that the case $n_{\text{pl}} = 0$, $n_{\text{s}} = 3$ and $\eta = 0.5$ is not a valid case for Eq. (2), as a single, diagonally polarized light beam cannot be thought of as two independent sources of electromagnetic fields. Having said that, we would like to point out that the comparison between the height of the $P(0)$ and $P(1)$ bars is not a good metric for defining whether we have a thermal or a non-thermal distribution. This is because, as pointed out by the referee, for small values of the mean photon number, even the limiting cases of thermal and coherent (Poissonian) distributions look alike, see Fig. R2 below. Therefore, to unequivocally distinguish between thermal and non-thermal distributions one needs to compute a mean-photon-number-independent metric, namely the zero-delay second-order correlation parameter: $g^{(2)} = 1 + ((\langle (\Delta \hat{n})^2 \rangle) - \langle \hat{n} \rangle^2) / \langle \hat{n} \rangle^2$, which for a thermal distribution is equal to 2, whereas for a coherent distribution is 1. Please note that in the case of $n_{\text{pl}} = 0$ and $\eta = 0.5$, if valid for Eq. (2), it would lead to a $g^{(2)} = 1.5$, which is not what is found experimentally in Fig. R1 and Fig. R3. What we observe in the experiment is that when blocking one of the slits, the light coming out of the unblocked slit does remain thermal, that is, it shows a $g^{(2)} \approx 2$. This result supports our central claim, which states that it is only when a plasmon---i.e., a second field source---is excited that photon statistics of the initially thermal field are modified. Then, as discussed above, it is in this case in which Eq. (2) describes our experiment.

Figure R2: Photon-number distribution for thermal (left column) and coherent (right column) light, considering different mean photon numbers: (a,b) $\bar{n} = 3$, (c,d) $\bar{n} = 2$, (e,f) $\bar{n} = 1$, and (g,h) $\bar{n} = 0.1$. Note that even though the mean photon number changes the bar heights, the $g^{(2)}$ remains the same for each type of source. Also note that even in the extreme case of low mean photon numbers, where both photon distributions are essentially the same, $g^{(2)}$ allows us to unequivocally distinguish between the two light sources. Finally, we point out that the $g^{(2)}$ values in (a) and (c) are not exactly 2, because the photon number is truncated at $N=30$. We can get the exact $g^{(2)} = 2$ by increasing the maximum value of N .

Furthermore, as shown in the updated Fig. R3, the traversed state indeed has thermal statistics with higher mean photon numbers.

Figure R3: Histograms displaying theoretical and experimental photon-number probability distributions of the output from either slit. In **a** and **c**, the second slit, where the SPPs are transmitted, is blocked. Therefore, the probability distribution corresponds to the transmitted thermal beam. In **b** and **d**, the first slit is blocked, thus the probability distribution represents the quantum statistics of the plasmonic mode. Note that the theoretical prediction belongs to the analytical expression for a single-mode thermal field and not that described by Eq. (2) of the main text.

Comment 4: On Reply to Comment 7: The authors have provide error bars and this has improved the manuscript considerably.

On Reply to (the final) Comment 8: It is useful that the authors provided the transmission coefficient of the first slit and the photon-to-pasmon efficiency in the second slit. And I agree with the authors' response regarding the photon-to-plasmon conversion efficiency: one could express Eq. (2) in terms of this efficiency but one can also leave it out as the authors did. This was enlightening.

Reply to Comment 4: We thank the reviewer for the positive feedback on our revised manuscript.

Comment 5: In Summary: All in all, I am quite satisfied with most of the authors' responses, but as is clear from Points 5 and 6 above, I still find myself disagreeing with their central claim that the non-thermal photon distributions are due to plasmons (which then “will establish new paradigms in quantum plasmonics”). But because of this disagreement with the central claim, I cannot recommend to accept this work for publication in its present form. My reasoning can be summarized as follows: I did a simple check that Eq. (1) gives the physically intuitive photon distributions also for very small (and even zero) average plasmon numbers, contrary to the statement by the authors that Eq. (1) stops being valid in that limit. In that limit of very small n_{pl} , plasmon effects are negligible but Eq. (2) still predicts non-thermal photon distributions $P_{det}(n)$, at least when choosing n_s large enough (and $n_s = 3$ is large enough). So that tells me that it is too quick to ascribe the non-thermal distributions as observed by the authors to plasmons. How could I be convinced by the authors? It would start here: it seems that the authors would like to use Eq. (2) for some values of n_{pl} but not for others, so I am asking them to provide a quantitative justification of the range of validity of their formula (2), i.e. for what values of n_{pl} it can be used and for what values it cannot. Such a justification at present is lacking.

Reply to Comment 5: We thank the reviewer for the multiple comments, undoubtedly these have allowed us to improve our manuscript. We believe that our reply to Comments 2 and 3 together with the additional experimental data therein will resolve the concerns raised by the reviewer.

Reviewers' Comments:

Reviewer #1:

None

Reviewer #3:

Remarks to the Author:

I have read the authors' replies and the revised manuscript.

It is now more clear that we all agree that Equation (1) is also valid for two sources even if one of them emits no photons. The important point is that Equation (2) does not describe the situations in the authors' experiments where only one slit is open. The authors have shown this convincingly with a series of control experiments. In Figure 3, I see that theory and experiment agree for all angles θ , but that there is an unexplained almost non-analytic jump from zero angle to the first measurement for the smallest nonzero polarization angle. I asked for an explanation of the limits of validity of formula (2). From their reply I understand that there is indeed a transition between the single-mode description for $\theta = 0$ to the curve described by their Eq. (2), but the theoretical explanation of this transition falls out of the scope of their work. I can agree with that, now that the theoretical and experimental situation for the case $\theta = 0$ have also been clarified in the main text and supplement. The authors' explanation that there is an unexplained transition region was very helpful for my understanding, since now I no longer think that there must be some kind of inconsistency in their explanation. Also helpful was the careful reply to my comment with new label 3, where the authors stress the importance of using a mean-photon-number-independent metric.

I also noticed that the authors agreed with the suggested changes by Referee 1.

In summary, I am satisfied with the response by the authors. I am convinced that they have done interesting and careful measurements of photon distributions and g_2 functions of double-slit experiments that involve plasmons, and that their measurements fit well with their theoretical explanation. I recommend publication in Nature Communications.